# *“Put Me in, Coach”*: A Discussion of Deprescribing Roles, Responsibilities, and Motivations Based on a Qualitative Study with Healthcare Professional Students

**DOI:** 10.3390/pharmacy13030077

**Published:** 2025-05-29

**Authors:** Devin Scott, Amy Hall, Rachel Barenie, Crystal Walker, Muneeza Khan, Paul Koltnow, William R. Callahan, Alina Cernasev

**Affiliations:** 1Department of Interprofessional Education, College of Graduate Health Sciences, University of Tennessee Health Science Center, Memphis, TN 38163, USA; dscott50@uthsc.edu; 2Teaching and Learning Center, University of Tennessee Health Science Center, Memphis, TN 38163, USA; 3Teaching and Learning Center, Department of Medical Education, College of Medicine, University of Tennessee Health Science Center, Memphis, TN 38163, USA; ahall32@uthsc.edu; 4Department of Clinical Pharmacy and Translational Science, College of Pharmacy, University of Tennessee Health Science Center, Memphis, TN 38163, USA; rbarenie@uthsc.edu; 5College of Nursing, University of Tennessee Health Science Center, Memphis, TN 38163, USA; cmarti47@uthsc.edu; 6Department of Family Medicine, College of Medicine, University of Tennessee Health Science Center, Memphis, TN 38163, USA; mkhan14@uthsc.edu; 7Physician Assistant Program, College of Medicine, University of Tennessee Health Science Center, Memphis, TN 38163, USA; pkoltnow@uthsc.edu; 8Department of General Dentistry, College of Dentistry, University of Tennessee Health Science Center, Memphis, TN 38163, USA; wcallaha@uthsc.edu

**Keywords:** deprescribing, deprescriptions, health education, polypharmacy, interprofessional education, qualitative research, healthcare team

## Abstract

As the US population ages, the number of prescriptions managed by patients and healthcare teams is increasing. Thus, discontinuing or reducing medications that are considered to pose more risks than benefits can be achieved through deprescribing. Despite increasing calls for a stronger focus on deprescribing in healthcare education, current discussions highlight the lack of training on this topic within healthcare curricula. This is a significant barrier to effectively implementing the deprescribing process. This study aimed to characterize healthcare professional students (HPSs)’s perspectives on deprescribing within an interprofessional healthcare team, particularly regarding the motivations and roles of these future practitioners. Methods: Focus groups were conducted with HPSs at the University of Tennessee Health Science Center. The data collection, guided by a conceptual model, took place over three months in 2022. Data analysis was performed using thematic analysis, during which themes were identified through inductive coding. Results: Participants (*n* = 36) represented various faculties, including medicine, pharmacy, health professions, nursing, and dentistry. Two themes emerged: (1) Healthcare Team Members’ Roles and Responsibilities (2) “*Put Me in, Coach*”: Patient Safety Motivates Deprescribing. Conclusion: Data from HPSs highlighted the importance of an interprofessional healthcare team approach to deprescribing. Based on these insights, educators and practitioners should focus on establishing strong interprofessional healthcare teams that privilege open communication. Teams should consider deprescribing as a patient safety concern, as this may galvanize the team and provide additional motivation for performing the necessary work of deprescribing.

## 1. Introduction

Deprescribing is a necessary and complex process through which members of a healthcare team initiate a plan of discontinuation or reduction in medications that are deemed to carry a higher risk than benefit [1]. The goal of deprescribing is reductions in polypharmacy to improve patient care and outcomes [1]. The benefits of deprescribing are well recognized, yet there are many barriers that interprofessional healthcare teams must overcome to successfully deprescribe [2].

Given the importance of deprescribing, education on deprescribing is vitally important. Indeed, recent research has advocated for more focused education on deprescribing in the healthcare curriculum [3]. Likewise, there have been repeated calls to increase the emphasis on deprescribing in the healthcare curriculum and to enact education standards that reflect the importance of deprescribing [4].

However, despite a growing chorus of calls for increased emphasis on deprescribing in healthcare education, contemporary work on deprescribing continues to call attention to the lack of education on deprescribing in the healthcare curriculum as a major barrier to successful initiation of the deprescribing process. In fact, research on prescribers has repeatedly identified the lack of deprescribing education as a barrier to deprescribing [1]. Similarly, a recurring barrier to student pharmacist initiation of deprescribing was identified to be a lack of education on deprescribing [5].

An important consideration when contemplating deprescribing education is a discussion of the roles and motivations of different members of the interprofessional healthcare team. Indeed, previous research has investigated student perceptions of their roles in deprescribing [6]. Other literature has touched on the motivations of trainees to initiate deprescribing [7]. That said, there is a need for a greater understanding of HPSs’ thoughts, experiences, and perceptions of the ways in which their roles and motivations inform the success of the interprofessional healthcare team in successfully initiating deprescribing. In this study, we endeavor to catalog and illuminate the perspectives of HPS trainees on deprescribing in the context of interprofessional healthcare teams, especially relating to the motivations and roles of these future practitioners.

## 2. Materials and Methods

This study used a qualitative approach consisting of focus groups (FGs) to gather insights about deprescribing in a healthcare interprofessional setting. The FGs were designed using the guidelines provided by Krueger and Krueger, 2002, which provide the necessary steps to ensure accuracy of the data collection [8]. Additionally, the FGs were influenced by a conceptual model of deprescribing by Linsky et al., 2019, which includes a formal patient, prescriber, and system factor steps that support a clinician in the process of initiating and discontinuing a medication [9]. This study was approved by the University of Tennessee Health Science Center (UTHSC) Institutional Review Board (IRB # 22-08592-XM approved on 3 February 2022).

Recruitment was focused on students enrolled in a healthcare professional degree program at UTHSC. An email was sent describing the study and inviting students to participate on a voluntary basis. HPSs confirmed their willingness to participate via email response. Confirmed participants were assigned to their FG based on scheduling availability, and FGs took place via Zoom. A total of 36 HPSs elected to take part in four, 90–120 min FGs, with a range of 6 to 11 participants per FG. There were no withdrawals and no non-participants involved in the FGs. Seventeen HPS had supervised clinical experiences, with nine fourth years and eight third years. After receiving a formal overview of exact study details, students gave verbal consent to continue. Students who chose not to consent were not penalized in any way and simply logged off the video call. Data collection continued until saturation was obtained. More details about the methodology can be found in Figure 1.

Figure 1: The flowchart presents the step-by-step research methodology used in this study. It emphasizes the models that informed the design and execution of focus groups and also outlines the data analysis procedures and key themes identified.

The FG was conducted by two Ph.D.-trained researchers with experience in qualitative studies, using an FG discussion guide. Each FG was audio-recorded and transcribed by a third party to avoid any bias during the transcription process. The corpus of data, including field notes taken by both Ph.D.-trained researchers, was analyzed inductively and deductively by three coders [AC, AH, DS] using the six steps outlined by Braun and Clarke 2010 ensuring a comprehensive understanding of the data [10]. Additionally, data were analyzed using Dedoose^®^, (Version 9.10.27, Los Angeles, CA, USA) a software for the analysis of qualitative (and mixed) data, which facilitated codes, sub-codes, categories, comments, and discussions, in addition to tracking contributors’ actions and changes. Three researchers further analyzed the data by developing the codes and categories necessary to determine any emergent themes. A previous manuscript details the methodology in more detail [11].

## 3. Results

### 3.1. Theme One: Healthcare Team Members’ Roles and Responsibilities

Theme one reflects student perspectives on the roles and responsibilities of members of the interprofessional healthcare team during the deprescribing process. Specifically, multiple students addressed which team member should be the one to initiate this process, which may be different than who should follow-up. The following quotes were selected to emphasize these viewpoints.

One medical student, FG1 P2, reflected on whose role it is to initiate the deprescribing process. She suggested that the physician or nurse practitioner should carry out the primary responsibilities, by saying “*it’s either going to be the MD or it could be the nurse practitioner, whoever is attending for that patient. And then, the RN after that is more just following the orders, so definitely the attending physician or nurse practitioner*.” [FG1 P2] 

FG4 P9 a pharmacy student from another FG echoed that sentiment. She states: “*Yeah, in my opinion, the final responsibility falls on the prescriber.*” [FG4 P9]

While this viewpoint appeared to be shared by other members of the FG, others believe the deprescribing process to be more collaborative. One pharmacy student stated “*I agree with FG1 P5. I think, at the end of the day, it is the prescriber’s authority that is going to decide that it should be deprescribed, but I think everyone plays a role in looking at that medication regimen, whether it’s missed or not by the prescriber, everyone should have a role in it.*” [FG1 P3]

Another student reiterated the collaborative nature of the healthcare team, emphasizing that each team member is part of a larger effort. “*So, I guess like the official authority to say no longer prescribe this would be like what FG1 P2 said, MD, the nurse practitioner, or physician assistant, but I do believe that sometimes people miss things and that anyone can help their team, whether it be pharmacy-- they know things that MDs don’t.*” [FG1 P5]

Furthermore, other students reflected on how the actual healthcare setting can impact who the responsibility of deprescribing falls on. One dentistry student explains ***“****I think that we do have a role in deprescribing, but with dentistry, we kind of place that responsibility on the physician, that’s what we’re taught time and time again because we don’t want to take that responsibility onto us, and so we kind of push it sideways.*” [FG3 P5] 

A pharmacy student also describes how certain scenarios can also be affected by scope of practice and involvement in the patient’s care. She emphasizes the unique knowledge and expertise they bring, highlighting the collective effort of the team members and their role in a unified and impactful healthcare team. “*I definitely think a good way to approach it is to make sure that you’re addressing it from your angle of expertise. So, as pharmacy, like drug–drug interactions obviously are at my forefront whereas maybe a med student may be looking along the lines of… how is your quality of life… but just making sure that we approach it from our unique specific angle to make sure that it’s getting addressed from all different aspects.*” [FG1 P6]

HPSs asserted that the primary responsibility of initiating deprescribing rests on the prescribing provider, whether this be the physician or the nurse practitioner. If, for some reason, this process is initiated by another member of the healthcare team, it was made clear that the physician (or equivalent prescriber) should make the ultimate decision. However, other students suggested that the initiation and final recommendation of deprescribing is a shared effort amongst all healthcare team members. While this responsibility can become unclear based on practice setting or scope of practice of the team member, many students believed that everyone on the healthcare team should play a role in the deprescribing process.

### 3.2. Theme 2: “Put Me in, Coach”: Patient Safety Motivates Deprescribing

The second theme emphasizes HPS’s perspective on the deprescribing process as one that prioritizes patient safety. The focus groups highlighted the necessity of collaboration, as various members of the healthcare team possess different areas of expertise for a reason. The following quotes highlight the importance of patient safety, which is a critical aspect of the patient care process.

One pharmacy student argued that deprescribing is a necessary process due to its potential for many positive outcomes. They also recognized the crucial role of pharmacists in this process, primarily in patient safety. For example, pharmacists might use their judgment and decide to not fill a prescription to ensure patient safety is achieved.

“*Another more common and easier-for-us-pharmacists method is if we catch an error or something we think is causing more harm than good, we call the prescriber and ask them is there something we’re missing… And if we think it will do harm, we can have our last resort of not filling it.*” [FG1 P1]

Additionally, another pharmacy student, FG3 P1, wrapped up their FG’s discussion by emphasizing that the strong motivation for her future role in this process is to ensure the patient’s health and safety is obtained. “*So I think what motivates me at the end of the day is the patient, talking to them and seeing their situation. Like if they tried this new, for example, an ACE inhibitor and they’re coughing, and they’re like, ugh, you know, you’re obviously going to reach out to the doctor, but seeing the patient like firsthand and making that discovery yourself, it’s like, oh, let me reach out to the prescriber, I can do that for you. Like just put me in, coach, like it’s me. You know, so it’s kind of knowing your role and being excited to do what you do every day I think is a great place to start.*” [FG3 P2] This participant also highlights the necessity of interprofessional collaboration. By asking the “coach,” or team leader, to “put them in,” or bring them into the deprescribing process, they call attention to both the interprofessional healthcare team and their unique role in deprescribing.

While many students made it clear that interprofessional collaboration is crucial in the deprescribing process, others highlighted the value of more interprofessional collaboration within their immediate circle of colleagues. FG3 P5, the dentistry student, indicated that the positive outcomes that can come from this, saying “*I think consulting with other dentists in our own building and also consulting with other physicians or specifically the physician and just running the idea by them and seeing what they think… that’s motivation to [deprescribe], but it has to be in a safe space more so when you go outside of your department.*” [FG3 P5]

Similarly, a medical student described “*So in my surgical setting, the biggest collaboration is between us surgeons and the anesthesiologist, where a patient might have better outcomes if they have a presurgical nerve block combined with enhanced recovery after anesthesia protocol.*” [FG2 P9]

In this theme, HPS repeatedly called attention to the importance of deprescribing. Although they emphasized various reasons for the importance of interprofessional collaboration in patient care, the main motivating factor identified was patient safety. Furthermore, they recognized this to be the rationale that will be utilized for initiating deprescribing in their future careers. Remaining in line with other findings, this was found to be the primary viewpoint across multiple disciplines.

## 4. Discussion

During focus group sessions, HPS discussed the roles and responsibilities of healthcare team members in the deprescribing process and their motivations for initiating this process. In the first theme, participants identified the prescriber, whether medical doctor, dentist, or nurse practitioner, as having the ultimate responsibility to deprescribe medications. However, participants recognized all members of the healthcare team as instrumental in the process through collaboration with the interprofessional healthcare team. Each member of the interprofessional team was said to contribute a unique knowledge and skill set that could ensure successful initiation and implementation of deprescribing via monitoring medications and voicing concerns when they arise. The second theme revealed participants’ motivations for initiating the deprescribing process, which included reducing patient harm and improving patient outcomes. Woven throughout this second theme was the reiteration of the importance the interprofessional team’s collaborative nature plays in the deprescribing process.

The first theme centered on the role and responsibility of the various members of the healthcare team in the deprescribing process. Overall, the participants agreed that the initial prescriber, whether a medical doctor, dentist, or nurse practitioner, held the ultimate responsibility for deprescribing. While other members of the healthcare team could initiate and contribute to the deprescribing conversation, the prescriber possessed final authority. This finding is consistent with practices surrounding prescribing and deprescribing medications. Additionally, this theme aligns with published literature which identifies the prescriber as responsible for deprescribing. In a qualitative study by Gerlach et al., the professional roles of general practitioners, community pharmacists, and nurses in deprescribing were explored via focus groups [12]. From these discussions, general practitioners were identified as the main authority in deprescribing [12].

The HPS students also identified other members of the healthcare team, e.g., nurses, pharmacists, dentists, etc., as holding responsibility for deprescribing. While they do not have final authority in this process, each has expertise that can aid in determining if and when deprescribing should occur. Participants indicated that all members of the healthcare team offer a unique perspective that can contribute to providing optimal care. This idea of an interprofessional approach to deprescribing is consistent with the published literature. Several studies highlight the importance of relying on the expertise of the interprofessional team when deprescribing [13,14,15]. During focus groups with physicians, nurses, and pharmacists, Foley et al. found that interprofessional collaboration and communication were key to providing patient care including deprescribing [13]. From the authors’ insights, “nurses highlighted the usefulness of having a pharmacist within their structure and the complementary benefit of mutual knowledge, which facilitates a reflective process on how medication is used” (p. 9) [13]. Furthermore, Radcliffe et al., in their review, postulated that deprescribing can be more successful when interprofessional teams collaborate and capitalize on each other’s strengths [14].

Patient care settings, e.g., outpatient, inpatient, dentist office, etc., can affect the deprescribing process and dictate who initiates and finalizes deprescribing. In line with the setting influencing deprescribing, the practitioner’s scope of practice or involvement in patient care can also impact if and how deprescribing occurs. Keller et al. interviewed hospitalists and nurses providing inpatient care to patients who had been prescribed benzodiazepines and sedative hypnotics by other physicians [16]. While these hospitalists have authority to deprescribe, they saw their role in patient care to be focused on stabilization and monitoring rather than deprescribing medications prescribed by an outpatient provider or educating the patient about risks of those medications [16]. This is somewhat contradictory to the HPSs in this study, who stated that all team members should monitor medications and address issues when seen. However, the idea of different settings and scopes of practice playing a role in the deprescribing process does align with the findings of this study.

The HPSs expressed their motivations for initiating deprescribing even when they did not hold ultimate authority as solely focused on eliminating patient harm and improving patient outcomes. Participants’ motivations for deprescribing focused on identifying potential medications where the risks outweighed the benefits and could cause more harm than good. Embedded in these motivations was the idea that all members of the healthcare team play an essential role in identifying these risks and initiating discussions about potential deprescribing. The participants reiterated the importance of viewing the patient through their unique lens and sharing this expertise with the interprofessional team.

Reducing patient harm and improving patient outcomes is the hallmark motivation to deprescribe in healthcare teams [17]. Scott et al., 2015 defined deprescribing as the “systematic process of identifying and discontinuing drugs in instances in which existing or potential harms outweigh existing or potential benefits” (p. 827) [17]. Based on this definition the health professional students’ motivations directly align with the purpose of deprescribing. In a cross-sectional survey, Goyal et al., 2020 identified that clinicians felt comfortable deprescribing when there was a high risk of an adverse drug event which was highlighted as the most reported indication for deprescribing cardiovascular medication [18]. Likewise, practitioner interviews by Keller et al., 2023 highlighted reducing patient harm as a motivator for deprescribing [16]. Pharmacists reported that “when they viewed a significant safety concern, such as the potential for accidental overdose, a high risk of falls, or concerns about renal problems, pharmacists also expressed feeling confident in reaching out to physicians to reduce the dose” (p. 5) [16].

While the interprofessional healthcare team has been found to be key in deprescribing, pharmacists’ roles in this process, as identified by physicians and nurses, are crucial to the success of this collaborative effort to reduce potential adverse drug events [13,14,15]. In a cross-sectional survey of 307 RNs by Ailabouni et al., 2019 “67.4 percent of the nurses agreed or strongly agreed that deprescribing implemented with the help of a clinical pharmacist would be beneficial” to the patients (p. 13) [19]. Similarly, in focus groups with physicians, nurses, and pharmacists, pharmacists were seen as crucial in reducing potential harms caused by medications [20]. Furthermore, a general practitioner surveyed by Gerlach et al.2020 noted the importance of pharmacists to the deprescribing process [12]. He stated that while he holds ultimate responsibility for deprescribing, he “need[s] the pharmacist to be a second line” (p. 5) of defense and speak up when he makes a mistake that might cause patient harm [12].

Reducing potential harm and improving patient outcomes, motivating factors for deprescribing identified in this study, can be successfully accomplished via interprofessional teamwork/collaboration [21]. The HPSs highlighted the importance of improving patient outcomes through a collaborative effort to deprescribe, which aligns with published literature [22]. In a randomized controlled trial, Balsom et al. found that patient outcomes could improve and potential harms could decrease when an interprofessional team, including the pharmacist, physician, nurses, patient, and family, collaborated to deprescribe medications [22]. Likewise, in a preliminary prospective observational study of older patients and their drug intake, interprofessional teamwork to deprescribe was found to “improve the risk/benefit ratio” of those patients’ therapies (p. A257) [23]. Finally, during focus groups with physicians, nurses, and pharmacists, all confirmed the necessity of teamwork to improve patient outcomes when deprescribing [20].

### Strength, Limitations, and Future Research

This study highlights the evidence-based approach to conducting qualitative research. It demonstrates the utility of focus groups in uncovering HPS perspectives on deprescribing, based on the training they have received at this point in their studies. Additionally, these reflections may serve as a guiding viewpoint in future formation of interprofessional collaborations to address deprescribing and improve the respective curricula. Although the opinions of those who participated in the focus groups may not be generalizable to other states or regions, this research provides a vehicle for developing the appropriate interprofessional approaches to better prepare HPS to act on deprescribing while in practice. One limitation is that, due to the self-professed lack of education about deprescribing indicated by some students, the students’ understanding of the concept and process of deprescribing may not have been at the appropriate competence level for an in-depth discussion. That said, the lack of education surrounding deprescribing is an important finding to report.

## 5. Conclusions

Findings with HPSs highlighted the importance of an interprofessional healthcare team approach to deprescribing. Furthermore, based on these discussions with future healthcare professionals, patient safety is a major motivator for initiation of and follow-through in deprescribing. Based on these insights, educators and practitioners interested in deprescribing should focus on establishing strong interprofessional healthcare teams that privilege open communication. At the same time, educators and members of the interprofessional healthcare team should consider framing deprescribing as a patient safety concern, as this may serve to galvanize the team and provide additional motivation for performing the necessary work of deprescribing. Ultimately, this study has practical implications for deprescribing educators and practitioners across the healthcare field. Practitioners would do well to establish and maintain a robust interprofessional healthcare team that can work together to deprescribe, while prioritizing patient safety. Deprescription educators should focus on leveraging patient safety to motivate HPSs to initiate deprescribing, while simultaneously engaging in educational activities, like interprofessional education, which strengthens students’ commitment to the interprofessional healthcare team.

## Figures and Tables

**Figure 1 pharmacy-13-00077-f001:**
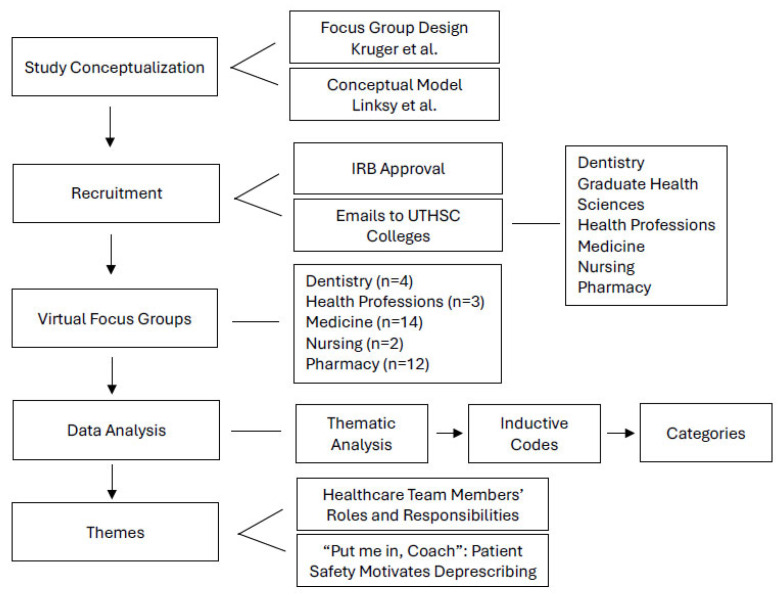
A synopsis of the study methodology and results [8,9].

## Data Availability

No new data were created or analyzed in this study. Data sharing is not applicable to this article.

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
