# Peer review of "“Put Me in, Coach”: A Discussion of Deprescribing Roles, Responsibilities, and Motivations Based on a Qualitative Study with Healthcare Professional Students"

_pharmacy, 2025, doi:10.3390/pharmacy13030077_

Round 1
Reviewer 1 Report
Comments and Suggestions for Authors
The submitted article titled "Put Me in Coach: A Discussion of Deprescribing Roles, Responsibilities, and Motivations Based on a Qualitative Study with Healthcare Professional Students" by Devin Scott et al. addresses a health-related important question. The authors discussed the deprescribing roles and responsibilities with healthcare professional students. Two main themes were highlighted after performing the focus group methods: healthcare Team Member's Roles and Responsabilities and Put me in Coach. Although this research paper has a high quality in my opinion, I believe the authors can improve it further.
- The study was conducted among healthcare students without field experience. Therefore, I would like to know if the authors assessed whether the students were comfortable with the topic and if it had been covered in their curriculum before their inclusion in the study.
- It seems that the authors employed a "constructive method", and it appears that they have already summarized the key themes. I wonder whether this approach might have introduced any biases into the students' topic development.
- The concept "Put Me in Coach" is highlighted by the authors. However, this theme seems not to be the primary outcome following the structure and the content of the article. In addition, it is still hard in the current version of the article to better understand this point. I would like to ask the authors to emphasize their description and discussion based on this point.
- I think the authors made a mistake as they presented the same figure twice. In addition, I would like to ask them to complete the legend of the figure.
- If possible, the authors can standardize the presentation of the references. Sometimes the original articles have their DOI, sometimes they don't.
Author Response
The submitted article titled "Put Me in Coach: A Discussion of Deprescribing Roles, Responsibilities, and Motivations Based on a Qualitative Study with Healthcare Professional Students" by Devin Scott et al. addresses a health-related important question. The authors discussed the deprescribing roles and responsibilities with healthcare professional students. Two main themes were highlighted after performing the focus group methods: healthcare Team Member's Roles and Responsabilities and Put me in Coach. Although this research paper has a high quality in my opinion, I believe the authors can improve it further.
- The study was conducted among healthcare students without field experience. Therefore, I would like to know if the authors assessed whether the students were comfortable with the topic and if it had been covered in their curriculum before their inclusion in the study.
Response: Thank you for this inquiry. The manuscript has been updated to reflect the number of HPS who have supervised clinical experiences. One interview guide question asked “What is Deprescribing?” Another asked “Are there any other definitions or things that you think of when you hear the term deprescribing.” Finally, we asked “What training would you say that you received on Deprescribing in your curriculum?” This is also addressed in “Strengths, Limitations, and Future Research” on page 8.
- It seems that the authors employed a "constructive method", and it appears that they have already summarized the key themes. I wonder whether this approach might have introduced any biases into the students' topic development.
Response: Thank you for the clarification. To prevent bias, the focus groups were conducted by an author who is not a clinical faculty member. Additionally, the transcription was performed by a third party. During focus group discussions, the interviewer did not introduce any themes to students. All themes were generated after the transcription data was analyzed, well after focus group discussions were conducted. Themes were constructed after thematic analysis of the qualitative data was conducted.
- The concept "Put Me in Coach" is highlighted by the authors. However, this theme seems not to be the primary outcome following the structure and the content of the article. In addition, it is still hard in the current version of the article to better understand this point. I would like to ask the authors to emphasize their description and discussion based on this point.
Response: Thank you for the valuable recommendation. We amended the text.
- I think the authors made a mistake as they presented the same figure twice. In addition, I would like to ask them to complete the legend of the figure.
Response: Thank you for this clarification. The figure now has a legend. Legend text: The flowchart presents the step-by-step research methodology used in this study. It emphasizes the models that informed the design and execution of focus groups, and also outlines the data analysis procedures and key themes identified.
- If possible, the authors can standardize the presentation of the references. Sometimes the original articles have their DOI, sometimes they don't.
Response: Thank you for this clarification. We used EndNote to address this issue.
Reviewer 2 Report
Comments and Suggestions for Authors
Dear authors,
I have read the manuscript pharmacy-3597102 thoroughly. Overall, I believe the quality of the text is more than sufficient for the paper to be published in “Pharmacy”, and may be of reasonably high interest for the readers of this magazine. I have to say that the level of English used throughout the manuscript is superb and enables the reader to follow the story without additional problem caused by so-often poor presentation.
Abstract gives all the necessary information about the contents of the paper, keywords are appropriately chosen and all the necessary literature is listed as required. There is no excessive self-citation by authors.
Introduction gives all the necessary information for the readers. I am very happy that the introduction is short and as such very reader-friendly.
The methods section gives sufficient information for the reader and the methodology of the research using discussion in focus groups is sound. Results are clearly presented. The same is true with Discussion, which is quite extensive and easy to follow. All parts are written in such a detail, that I have no further suggestions. The possible limitations of the study are also clearly presented.
I suggest that the manuscript is accepted for publication in present form.
Best regards,
Aleš Obreza
Author Response
I have read the manuscript pharmacy-3597102 thoroughly. Overall, I believe the quality of the text is more than sufficient for the paper to be published in “Pharmacy”, and may be of reasonably high interest for the readers of this magazine. I have to say that the level of English used throughout the manuscript is superb and enables the reader to follow the story without additional problem caused by so-often poor presentation.
Abstract gives all the necessary information about the contents of the paper, keywords are appropriately chosen and all the necessary literature is listed as required. There is no excessive self-citation by authors.
Introduction gives all the necessary information for the readers. I am very happy that the introduction is short and as such very reader-friendly.
The methods section gives sufficient information for the reader and the methodology of the research using discussion in focus groups is sound. Results are clearly presented. The same is true with Discussion, which is quite extensive and easy to follow. All parts are written in such a detail, that I have no further suggestions. The possible limitations of the study are also clearly presented.
I suggest that the manuscript is accepted for publication in present form.
Response: Thank you for your time to review our manuscript. We greatly appreciate it.
Reviewer 3 Report
Comments and Suggestions for Authors
Overall, the research is interesting and well-designed, but the manuscript needs further improvement, as it is somewhat sloppily written.
Figure 1 is included twice (on page 3 and again on page 4) and lacks a caption (legend).
Authors' affiliations include an abbreviation only (UTHSC) instead of the full institution name (see lines 7-20).
The second theme is purported to be described by the popular song's lyrics, is actually not well defined, and a reader can hardly understand its meaning unless he is familiar with John Fogerty's song. Moreover, the motto "Put me in, coach" is properly written in Figure 1 only, while it is misspelled in several other places (in the title where it reads "Put Me in Coach" and in lines 35 and 153).
Methods are not sufficiently described. Neither the number of focus groups (FGs) nor the number of participants in each FG is given. The previously published manuscript has been referred to, but that is not enough (BTW, what about duplicate/salami publication?). Please provide a better study description. Also, Dedoose® software is referred to as "a qualitative software" instead of "a software for the analysis of qualitative (and mixed) data".
Suggestions:
Some substantives are mentioned several dozen times (e.g. "healthcare professional students") while they could preferably be abbreviated. Please consider introducing an abbreviation instead of repeating them.
Check the keywords: Are the keywords taken from Medical Subject Headings (MeSH) or not?
For example, the MeSH descriptor is "Deprescriptions", and it is O.K. to use the word "Deprescribing" for a process, but the keyword might be one taken from MeSH.
Use Copy-Editing to distinguish quotes (just like in a previously published manuscript referred to as 11). Also, there are some duplications connected with quotations, e.g. FG1 P2 is mentioned twice (in line 114 and again in line 118), as well as the next one, FG4 P9 (in line 118 and in line 120), and a comma is missing between the two of them in line 118.
Author Response
Overall, the research is interesting and well-designed, but the manuscript needs further improvement, as it is somewhat sloppily written.
Figure 1 is included twice (on page 3 and again on page 4) and lacks a caption (legend).
Response: Thank you for this suggestion. It was addressed.
Authors' affiliations include an abbreviation only (UTHSC) instead of the full institution name (see lines 7-20).
Response: Thank you for this suggestion. We removed the UTHSC abbreviation.
The second theme is purported to be described by the popular song's lyrics, is actually not well defined, and a reader can hardly understand its meaning unless he is familiar with John Fogerty's song. Moreover, the motto "Put me in, coach" is properly written in Figure 1 only, while it is misspelled in several other places (in the title where it reads "Put Me in Coach" and in lines 35 and 153).
Response: Thank you for this clarification. We amended the text so that readers will not need to be familiar with the song. Additionally, we have corrected the grammar throughout the text.
Methods are not sufficiently described. Neither the number of focus groups (FGs) nor the number of participants in each FG is given. The previously published manuscript has been referred to, but that is not enough (BTW, what about duplicate/salami publication?). Please provide a better study description. Also, Dedoose® software is referred to as "a qualitative software" instead of "a software for the analysis of qualitative (and mixed) data".
Response: Thank you for this valuable suggestion. We amended the text.
Suggestions:
Some substantives are mentioned several dozen times (e.g. "healthcare professional students") while they could preferably be abbreviated. Please consider introducing an abbreviation instead of repeating them.
Response: Thank you for this valuable suggestion to use an abbreviation for “healthcare professional students” (HPS) which was used consistently throughout the manuscript.
Check the keywords: Are the keywords taken from Medical Subject Headings (MeSH) or not?
For example, the MeSH descriptor is "Deprescriptions", and it is O.K. to use the word "Deprescribing" for a process, but the keyword might be one taken from MeSH.
Response: Thank you for this recommendation. Deprescribing was found as a MeSH term here: http://id.nlm.nih.gov/mesh/M000605486. Deprescriptions have been added as an additional keyword. All other terms were amended to match MeSH keywords.
Reviewer 4 Report
Comments and Suggestions for Authors
The study is well-structured, with clear background, objectives, methods, results, and conclusions.
Minor comments: were there any session guide questions to guide focus group discussion?
Authors didn't discuss any barriers of deprescribing. Discussiong these barriers will enhance the manuscript.
Also, the role after during (such as communication) and after deprescribing (follow up and monitoring) should be discussed.
Author Response
The study is well-structured, with clear background, objectives, methods, results, and conclusions.
Minor comments: were there any session guide questions to guide focus group discussion?
Response: There was a focus group discussion guide that researchers followed when conducting focus group discussions. This guide is not included to protect proprietary data.
Authors didn't discuss any barriers of deprescribing. Discussing these barriers will enhance the manuscript.
Response: Unfortunately, a discussion of barriers was outside the scope of the themes discussed in this manuscript. The themes present in the data were centered on motivations and roles/responsibilities. Two themes emerged:1) Healthcare team members’ Roles and Responsibilities 2) “Put Me in, Coach”: Patient Safety Motivates Deprescribing.
Also, the role after during (such as communication) and after deprescribing (follow up and monitoring) should be discussed.
Response: Thank you for this insight. Unfortunately, no themes or subthemes emerged from the focus group discussions that could be included in this manuscript. We agree that future research should address roles related to communication, follow up, and monitoring.